# Impairment in quantitative microvascular function in non-ischemic cardiomyopathy as demonstrated using cardiovascular magnetic resonance

**Jeremy A. Slivnick**[1]*, **Karolina M. Zareba**[2], **Vien T. Truong**[3], **Ellen Liu**[2], **Alexis Barnes**[4], **Wojciech Mazur**[3], **Philip Binkley**[2]

1 Division of Cardiovascular Medicine, University of Chicago Medicine, Chicago, IL, United States of America, 2 Division of Cardiovascular Medicine, The Ohio State University Wexner Medical Center, Columbus, OH, United States of America, 3 Division of Cardiology, The Christ Hospital Health Network, Cincinnati, OH, United States of America, 4 Division of Cardiology, Northwestern Medicine, Chicago, IL, United States of America

* Jeremy.slivnick@uchospitals.edu

**Data Availability Statement:** Data cannot be shared publicly as this study was approved by the Ohio State University institutional review board

## Abstract

### Background

Microvascular dysfunction (MVD) is present in various cardiovascular diseases and portends worse outcomes. We assessed the prevalence of MVD in patients with non-ischemic cardiomyopathy (NICM) as compared to subjects with preserved ejection fraction (EF) using stress cardiovascular magnetic resonance (CMR).

### Methods

We retrospectively studied consecutive patients with NICM and 58 subjects with preserved left ventricular (LV) EF who underwent stress CMR between 2011–2016. MVD was defined visually as presence of a subendocardial perfusion defect and semiquantitatively by myocardial perfusion reserve index (MPRI<1.51). MPRI was compared between groups using univariate analysis and multivariable linear regression.

### Results

In total, 41 patients with NICM (mean age 51 ± 14, 59% male) and 58 subjects with preserved LVEF (mean age 51 ± 13, 31% male) were identified. In the NICM group, MVD was present in 23 (56%) and 11 (27%) by semiquantitative and visual evaluation respectively. Compared to those with preserved LVEF, NICM patients had lower rest slope (3.9 vs 4.9, p = 0.05) and stress perfusion slope (8.8 vs 11.7, p<0.001), and MPRI (1.41 vs 1.74, p = 0.02). MPRI remained associated with NICM after controlling for age, gender, hypertension, ethnicity, diabetes, and late gadolinium enhancement (log MPR, β coefficient = -0.19, p = 0.007).

(IRB) whom did not approve such public sharing of data. Data can be made available from the Ohio State University MRI Research team (contact: Suzanne Smart, Suzanne.Smart@osumc.edu) for researchers who meet the criteria for access to confidential data.

**Funding:** The author(s) received no specific funding for this work.

**Competing interests:** The authors have declared that no competing interests exist.

**Abbreviations:** AIF, arterial input function; CAD, coronary artery disease; Cardiac Magnetic Resonance, CMR; CFR, coronary flow reserve; ICA, invasive coronary angiography; LGE, late gadolinium enhancement; LVEF, left ventricular ejection fraction; MACE, major adverse cardiovascular events; MVD, microvascular disease; MPRI, myocardial perfusion reserve index; NICM, Non-ischemic Cardiomyopathy; NYHA, New York Heart Association; PET, positron emission tomography; PSIR, phase sensitive inversion recovery.

## Conclusions

MVD—as assessed using CMR—is highly prevalent in NICM as compared to subjects with preserved LVEF even after controlling for covariates. Semiquantitative is able to detect a greater number of incidences of MVD compared to visual methods alone. Further studies are needed to determine whether treatment of MVD is beneficial in NICM.

## Introduction

Coronary microvascular disease (MVD)—defined as impaired augmentation of coronary circulation with ischemia in response to stress in the absence of epicardial coronary artery disease (CAD)—afflicts a significant proportion of patients across a range of cardiovascular disorders and is associated with adverse outcomes [1, 2]. In a healthy individual, vasodilation of small vessel arterioles in response to stress allows for augmentation of the coronary microcirculation by as much as 4–5 times its original value [3]. This process is adaptive and allows for the delivery of increased coronary blood flow to meet increased demand in response to physiologic stress. However, stress-induced microcirculatory response may be reduced by more than half in the presence of MVD. The pathophysiology of MVD is thought to be multifactorial and related to small vessel atherosclerosis, vascular rarefaction, and endothelial dysfunction [4].

MVD can be assessed using invasive and non-invasive techniques. Microvascular reactivity can be evaluated with invasive coronary angiography (ICA) using coronary flow reserve (CFR) or the index of myocardial resistance (IMR) in response to a stress agent [5]. However, these invasive methods harbor procedural risks. Non-invasively, microvascular perfusion reserve can be assessed with positron emission tomography (PET) or cardiovascular magnetic resonance (CMR). CMR takes advantage of first pass gadolinium perfusion imaging in order to quantify semi-quantitative or fully quantitative changes in perfusion in response to a stress agent—typically adenosine. CMR-derived myocardial perfusion reserve index (MPRI) has been shown to correlate with CFR on ICA [6].

MVD is increasingly recognized as a contributor to symptoms and poor outcomes in a variety of disease states including hypertension, dyslipidemia, diabetes, and in chest pain syndromes in the absence of epicardial coronary artery stenosis [7–12]. MVD has also long been thought to contribute to the pathogenesis and progression of nonischemic dilated cardiomyopathy (NIDCM), but its presence has not been robustly confirmed in human subjects. Data from animal models have demonstrated its presence and clinical studies have implied its role in disease progression and symptoms in patients with NIDCM [13, 14]. However, wide scale investigation of MVD in humans with heart failure has awaited noninvasive techniques such as CMR. While Previous studies have been performed using PET, much less is known about the capabilities of CMR to assess MVD in NICM [2]. An added advantage to CMR over other noninvasive techniques is its tissue characterization properties which allow for the evaluation of many causes of cardiomyopathy including ischemic, valvular, infiltrative, and infectious processes. For this reason, CMR is frequently and appropriately obtained in the evaluation of new systolic heart failure [15]. CMR is therefore an ideal noninvasive modality to assess the microvasculature in this patient population.

We therefore performed this retrospective analysis to determine the prevalence of MVD—as derived using stress CMR—in patients with NICM as compared to subjects with preserved left ventricular ejection fraction (LVEF) referred for clinical CMR examination. We additionally wished to identify clinical and CMR-based differences between those NICM patients with and without impaired microvascular function.

## Materials and methods

### Study participants

We retrospectively identified 41 patients with NICM and 58 subjects with preserved LVEF and without clinical heart failure who had undergone comprehensive adenosine stress CMR perfusion between 2011 and 2016 at a single academic institution. NICM was defined in accordance with guidelines by a clinical diagnosis of heart failure with LVEF ≤50% requiring loop diuretics and free of obstructive CAD (defined by the presence of >50% coronary artery stenosis on invasive or CT coronary angiography, prior myocardial infarction, prior percutaneous or surgical revascularization, or inducible epicardial ischemia) [16]. The preserved LVEF cohort was identified using consecutive patients referred for stress CMR examination between 2011 and 2016 free of heart failure, obstructive CAD, prior myocardial infarction, percutaneous or surgical revascularization. Those with a clinical diagnosis of heart failure—including heart failure with preserved ejection fraction, hypertrophic and/or infiltrative cardiomyopathies—and those requiring loop diuretics were excluded from the preserved LVEF group. Those with evidence of infarct scar (i.e. subendocardial or transmural) on CMR and those with severe valvular regurgitation or stenosis were also excluded from both cohorts. Using similar methodology to a recent study by Zhou *et al*, patients found to have a regional (i.e. non-circumferential) perfusion abnormality on stress CMR were excluded from both cohorts unless obstructive coronary artery disease was subsequently excluded by invasive or CT coronary angiography [17]. Additionally, those with disease-specific non-ischemic LGE patterns—such as myocarditis, sarcoidosis, or arrhythmogenic cardiomyopathy—were excluded from the preserved LVEF cohort. In the preserved LVEF cohort, the indication for stress CMR was chest pain in 38 (66%), ventricular ectopy or ventricular tachycardia in 7 (12%), dyspnea in 3 (5%), syncope in 2 (3%), and other in 8 (14%) patients.

Clinical patient characteristics and comorbidities were established through a review of the electronic medical record. The following baseline clinical characteristics were collected: age, gender, ethnicity, body mass index, New York Heart Association Class, NICM etiology, serum creatinine, hematocrit, BNP, and troponin. The Ohio State University Institutional Review Board approved this retrospective study and agreed to waive informed consent. As this study was retrospective, patients were not directly involved in the research process. The authors of this manuscript have agreed to make the de-identified data available upon request.

### CMR imaging and analysis

Patients underwent clinical CMR exams using a 1.5 Tesla scanner (Magnetom Avanto or Espree, Siemens Medical Solutions, Erlangen, Germany). LV volumes, mass, and EF were assessed using steady state free precession (SSFP) sequences. Ventricular volumes and function were quantified from endocardial and epicardial tracing of serial short axis slices at end diastole and end systole. LV mass was calculated by multiplying the total myocardial volume at end diastole by the specific gravity of the myocardium (1.05 g/ml) [18].

Vasodilator stress CMR was performed using a 140 mcg/kg/min adenosine infusion for 2 minutes prior to first-pass perfusion imaging, and continued until completion of the perfusion imaging data acquisition. First-pass perfusion imaging was performed in 3 short axis slices using a 0.05 mmol/kg bolus of gadolinium. The mid-ventricular slice was chosen for analysis to maintain consistency and minimize artifact related to partial volume effects. A rest perfusion study was performed using the same protocol. Myocardial perfusion defects were assessed by both visual and semiquantitative analysis. Quantification was performed by manually delineating endocardial and epicardial left ventricular borders in the mid-short axis slice during

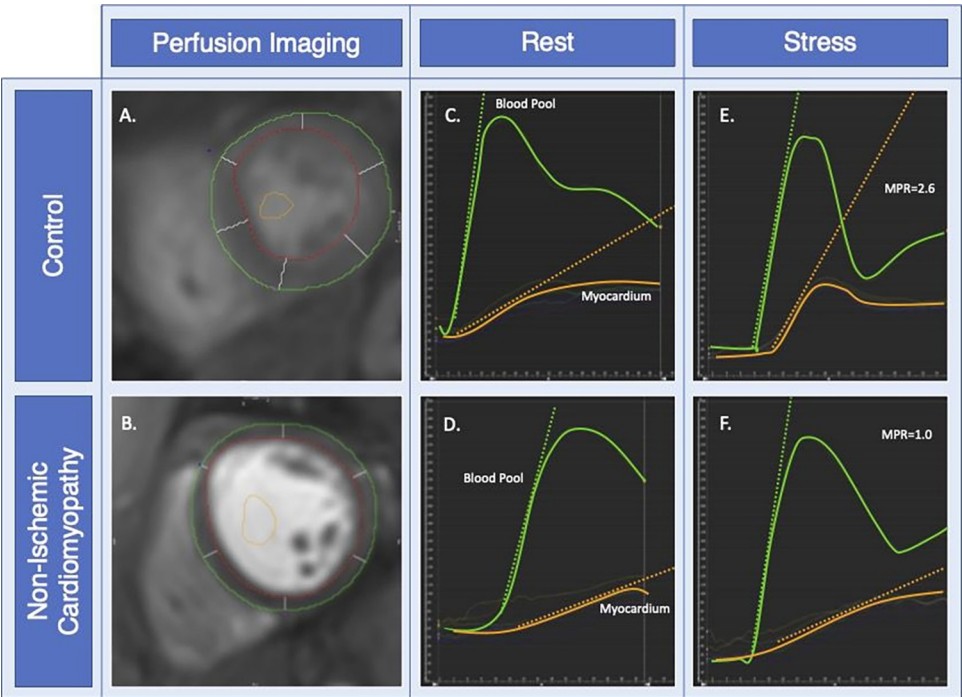

**Fig 1. Determining myocardial perfusion reserve index.** Panel A, B: First pass perfusion images with endocardial, epicardial, and blood pool contours. Panel C-F: Time intensity curve graphs at rest (C, D) and stress (E, F) first pass perfusion with maximal upslopes of blood pool (orange line) and myocardium (blue line). MPRI is calculated as the ratio of RUstress/RUrest where RU is ratio of maximal upslope of myocardium divided by blood pool.

both stress and rest first-pass perfusion with care to exclude blood pool activity (Cvi42, Circle Cardiovascular Imaging, Calgary, Canada). All cardiac borders were traced and subsequently independently verified by a level III CMR reader. Additionally, stress perfusion images were also closely reviewed by an advanced level 3 CMR reader to ensure that areas of dark rim artifact were not included within the tracings. Only mid-short axis slices were used out of concern for partial volume effects related to thin distal segments in the NICM group. Segments were not excluded based on the presence or absence of late gadolinium enhancement. An arterial input function (AIF) was traced in the blood pool to provide a reference with care to examine all slices to ensure there was no contamination with myocardium or papillary muscle tissue. Signal intensity curves of segmented myocardium were automatically generated from manually traced first pass perfusion slices (Fig 1). From these curves, relative upslope (RU) was automatically calculated both and rest and stress as the maximal myocardial upslope of myocardium divided by that of the blood pool using previously defined methods [19]. Patients in whom image quality was deemed inadequate for analysis by either reader were excluded. MPRI was defined as the ratio of RUstress/RUrest. MVD was defined quantitatively as MPRI <1.51 which was the lower interquartile range for the entire cohort and is similar to that used in the WISE subanalysis [20]. Qualitative MVD was defined as the presence of a circumferential subendocardial perfusion defect on first pass stress imaging [19]. Per guidelines, defects which occurred prior to contrast arrival in the LV myocardium, persisted <10 heart beats, or were <2 pixels wide were considered to be due to dark rim artifact and were not identified as true perfusion defects [21].

Late gadolinium enhancement imaging was performed using gradient-echo inversion recovery sequences and phase sensitive inversion recovery (PSIR) reconstructions 10 minutes

after administration of an additional 0.1 mmol/kg of GBCA [22]. The presence of LGE was assessed by 2 expert level 3 trained operators blinded to clinical data and had to be present in either two consecutive short axis slices or in two orthogonal imaging planes. LGE was scored according to its presence and extent based on the number of American Heart Association segments [23].

## Statistical analysis

Categorical data are presented as frequency with percentage, and comparisons between groups were performed using the chi-square test or Fisher exact test. Skewness, kurtosis, and visual inspection of the histogram and QQ plot were checked to assess the distribution of continuous variables. Continuous variables are presented as mean ± standard deviation (SD) for normal distribution or expressed as median (interquartile range) for non-normal distribution. Continuous variables were compared using Student's paired t-test or the Wilcoxon rank-sum test, as appropriate. Univariate and multivariable linear regression was performed to assess the relationship between the presence of NICM and MPRI after controlling for significant covariates. Univariate linear regression was performed to assess the relationship between MPRI and LV end diastolic volumes, mass, LVEF, and wall thickness. Because of non-normal distributed residual in multivariable analysis, logarithmic transformation was performed. To test the robustness of association of non-ischemic cardiomyopathy and RPP, the bootstrap method with 2,000 resampling technique was performed to estimate 95% bias-corrected and accelerated confidence intervals. Further, gamma regression model with an identity link function was applied to assess the robustness of result. Regression diagnostics were performed to test model assumptions. Statistical analyses were performed using R software, version 4.03 (The R Foundation, Vienna, Austria).

## Results

### Demographics

In total, 41 patients with NICM (mean age 51 ± 14, 59% male) and 58 subjects with preserved LVEF (mean age 51 ± 13, 31% male) identified. Within the NICM group, the etiology of cardiomyopathy was secondary to drug/toxin, genetic, hypertension, myocarditis, sarcoidosis, other, and idiopathic in 8 (20%), 3 (7%), 2 (5%), 2 (5%), 2(5%), 5 (12%), and 19 (41%) respectively (**Fig 2**). Of the patients with NICM, 22 (54%) were NYHA Class I, 12 (29%) were Class II, and 7 (17%) belonged to Class III. No patients endorsed NYHA Class IV symptoms. Clinical characteristics of the NICM and subjects with preserved LVEF are presented in **Table 1**. Compared with subjects with preserved LVEF, patients with NICM were more likely to be male gender and of African American ethnicity (**Table 1**). There were no significant differences in rates of classic cardiovascular risk factors (diabetes, hypertension, hyperlipidemia) or atrial arrhythmias between groups. As compared to patients with preserved LVEF, those with NICM were more likely to be on goal directed heart failure medications (**S1 Table**).

### CMR characteristics

A comparison of CMR parameters is presented in **Table 1**. Compared with subjects with preserved LVEF, patients with NICM had significantly higher left atrial volumes, LV volumes and mass, and lower LVEF. Late gadolinium enhancement was significantly more prevalent in the NICM vs those with preserved LVEF (51% vs 16%, p<0.001). In all instances, LGE patterns were in a non-ischemic distribution—either mid-wall or epicardial—as those with LGE in a subendocardial or transmural infarct pattern were excluded. Among those in the preserved

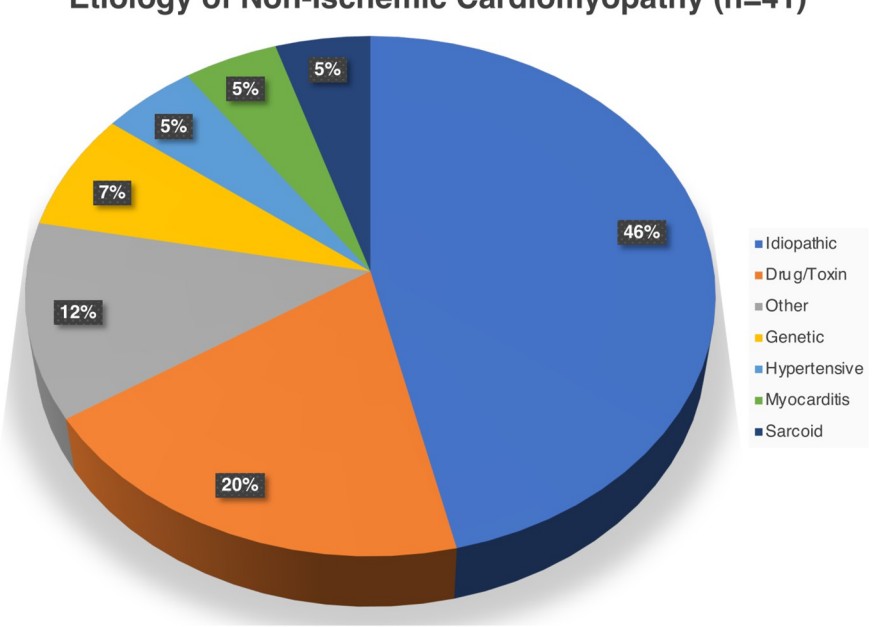

**Fig 2. Etiologies of non-ischemic cardiomyopathy for patients in the NICM cohort.**

LVEF cohort in whom LGE was present, LGE was located in the insertion point in 7 (78%) patients; the remaining 2 (22%) patients had faint, non-specific inferolateral LGE.

Only one patient in the NICM cohort had a segmental (i.e. non-circumferential) perfusion defect; in this patient, obstructive CAD was subsequently excluded by CT angiography. None of the patients within the preserved LVEF group had a segmental perfusion defect. First pass perfusion analysis revealed that patients with NICM has significantly lower MPR slopes both at rest (3.9 vs 4.9, p = 0.045) and with stress (8.8 vs 11.7, p<0.001) when compared to subjects with preserved LVEF. MPRI was significantly lower in the NICM group as compared with those with preserved LVEF (1.41vs 1.74, p = 0.02) (**Fig 3**). A density plot showing the distribution of MPRI for NICM and the preserved LVEF cohorts is displayed in **Fig 4**. In the NICM cohort, 23 (56%) patients had MVD by semiquantitative analysis (MPRI<1.51) while only 11 (27%) by visual analysis. In the preserved LVEF cohort, 14 (24%) patients had MVD by semiquantitative analysis while none were noted to have MVD by visual criteria. MPRI was not significantly associated with indexed LV mass (p = 0.18, $r^2$ = 0.03), indexed LV end diastolic volume (p = 0.18, $r^2$ = 0.16), LVEF (p = 0.11, $r^2$ = 0.03), septal (p = 0.99, $r^2$<0.0001), or posterior (p = 0.52, $r^2$ = 0.52) wall thickness.

## Comparison of normal versus impaired MPR in non-ischemic cardiomyopathy group

Amongst those with non-ischemic cardiomyopathy, there were 23 (56%) patients with MVD. A comparison of non-ischemic cardiomyopathy patients with and without impaired MPRI is presented in **Table 2**. Those with impaired MPRI were more likely to be hypertensive as compared to those with normal MPI. There was a trend towards increased prevalence of female gender amongst those with impaired MPRI. There were otherwise no significant intergroup differences with respect to comorbidities or CMR parameters.

**Table 1. Clinical and CMR characteristics in patients with non-ischemic cardiomyopathy vs the control cohort.**

| | | Preserved LVEF (N = 58) | NICM (N = 41) | P-value |
|---|---|---|---|---|
| **Clinical Characteristics** | | | | |
| Age (years) | | 51 ± 13 | 51 ± 14 | 0.95 |
| Gender, N (% male) | | 18 (31%) | 24 (59%) | 0.006 |
| Ethnicity | White, N (%) | 52 (90%) | 28 (68%) | 0.02 |
| | Black, N (%) | 4 (7%) | 12 (29%) | |
| | Hispanic, N (%) | 1 (2%) | 0 (0%) | |
| | Other, N (%) | 1 (2%) | 1 (3%) | |
| Diabetes, N (%) | | 7 (12%) | 6 (15%) | 0.77 |
| Hypertension, N (%) | | 25 (43%) | 24 (59%) | 0.16 |
| Hyperlipidemia, N (%) | | 21 (36%) | 16 (39%) | 0.83 |
| Atrial Fibrillation/Flutter, N (%) | | 9 (16%) | 3 (7%) | 0.35 |
| Creatinine (mg/dL) | | 0.90 ± 0.35 | 0.93 ± 0.23 | 0.32 |
| Hematocrit (%) | | 39±6 | 38±6 | 0.68 |
| B-type Natriuretic Peptide (ng/L) | | 63 (29–202) | 78 (39–991) | 0.39 |
| Troponin-I (ng/mL) | | 0.01 (0.01–0.02) | 0.01 (0.0–0.05) | 0.17 |
| **CMR Characteristics** | | | | |
| LVEDD (cm) | | 4.7 (4.4–5.1) | 5.6 (5.3–6.3) | <0.001 |
| LVEDV (mL) | | 122 (106–142) | 184 (154–217) | <0.001 |
| LVEDV Indexed (mL/m$^2$) | | 63 (58–75) | 97 (70–107) | <0.001 |
| LVESV (mL) | | 45 (37–53) | 107 (83–144) | <0.001 |
| LVESV Indexed (mL/m$^2$) | | 22.8 (20.0–28.0) | 55.2 (41.4–68.6) | <0.001 |
| LVEF (%) | | 64 ± 6 | 42 (32–45) | <0.001 |
| LA Volume Indexed (gm/m$^2$) | | 39 (30–48) | 45 (40–59) | 0.001 |
| LV mass (gm) | | 83 (64–102) | 110 (86–130) | <0.001 |
| LV mass Indexed (gm/m$^2$) | | 40.8 (35.3–48.9) | 55.2 (46.3–63.7) | <0.001 |
| LGE presence (%) | | 9 (16) | 21 (51) | < 0.001 |
| Number of segments | | 2 (1–3) | 3 (2–4) | 0.16 |
| Myocardial Perfusion Slope (rest) | | 4.9 (4.1–7.0) | 3.9 (3.2–5.3) | 0.045 |
| Myocardial Perfusion Slope (stress) | | 11.7 (8.8–16.0) | 8.8 (6.1–10.5) | <0.001 |
| Myocardial Perfusion Reserve Index | | 1.74 (1.51–2.09) | 1.41 (1.19–1.93) | 0.02 |

LV: left ventricle; EDD: end diastolic dimension; EDV: end diastolic volume; EF: ejection fraction; ESV: end systolic volume; LA: left atrium; LGE: late gadolinium enhancement.

## Multivariable analysis

In unadjusted analysis, compared to subjects with preserved LVEF, patients with non-ischemic cardiomyopathy had significantly lower MPRI (log MPR, β coefficient = -0.14, p = 0.03). After adjusting for age, gender, ethnicity, diabetes, hypertension, and presence of LGE, non-ische-mic cardiomyopathy remained independently associated with lower MPRI (log MPR, β coefficient = -0.17, p = 0.009) (**Table 3, Fig 4**). Bootstrap method and gamma regression confirmed that there was a significant association between non-ischemic cardiomyopathy and lower MPRI (**Table 3, S2 Table**). Additionally, the presence of hypertension was also independently associated with MVD.

## Discussion

We evaluated the prevalence of MVD in patients with NICM as compared to subjects with pre-served LV function using visual and semiquantitative CMR perfusion analysis. We found that

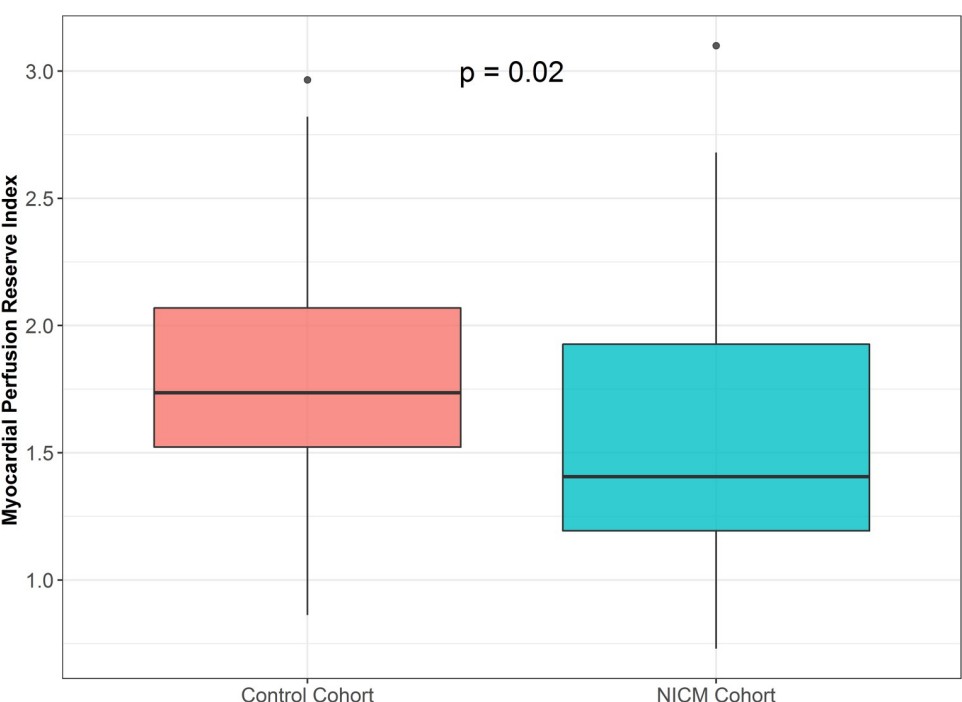

**Fig 3. Myocardial perfusion reserve index (MPRI) in the non-ischemic cardiomyopathy (NICM) vs the preserved LVEF cohorts.** Patients with NICM have significantly more impaired MPRI.

MVD as assessed using CMR is highly prevalent in NICM as compared to those with preserved LVEF. The key findings of the manuscript are summarized in the **S1 Graphical** abstract.

MVD is frequently identified in patients with NICM, consistent with findings in animal models and indirect evidence in clinical studies [13, 14]. The prevalence of MVD in patient with NICM was independents of other clinical variables indicating its unique contribution to disease pathophysiology. In our cohort of patients with NICM over half exhibited impaired myocardial perfusion reserve index. The prevalence of MVD in NICM was doubled as compared to subjects with preserved LVEF. This parallels findings from other studies which previously identified a high prevalence of MVD in other disease entities including hypertension, hyperlipidemia, diabetes, and in women with chest pain syndromes [7–11]. In our cohort, semiquantitative analysis identified a greater number of patients with MVD as compared with qualitative methods. This is consistent with prior CMR based studies which have identified MVD even in the absence of visual perfusion defects in patients with cardiac syndrome X [24]. Several potential explanations exist for these findings. Visual analysis relies on the detection of a subendocardial perfusion defect relative to the mid-myocardium and epicardium. This method may fail to detect MVD in cases where stress perfusion is impaired across all three myocardial layers. Additionally, circumferential perfusion defects may sometimes be difficult to differentiate from dark rim (Gibbs) artifact, a phenomenon in which a dark rim is artifactually observed at the interface between the brighter blood pool and darker myocardium.

We demonstrate a robust relationship between the presence of NICM and CMR-derived MPRI even after controlling for clinical and CMR covariates. Our study parallels a growing body of evidence including most notably Gulati *et al* in demonstrating a strong link between NICM and MVD [2, 25, 26]. A notable strength of the Gulati *et al* study was the utilization of fully quantitative CMR perfusion [26]. Fully quantitative perfusion has the potential to offer highly accurate quantification of myocardial blood flow at both stress and rest with robust

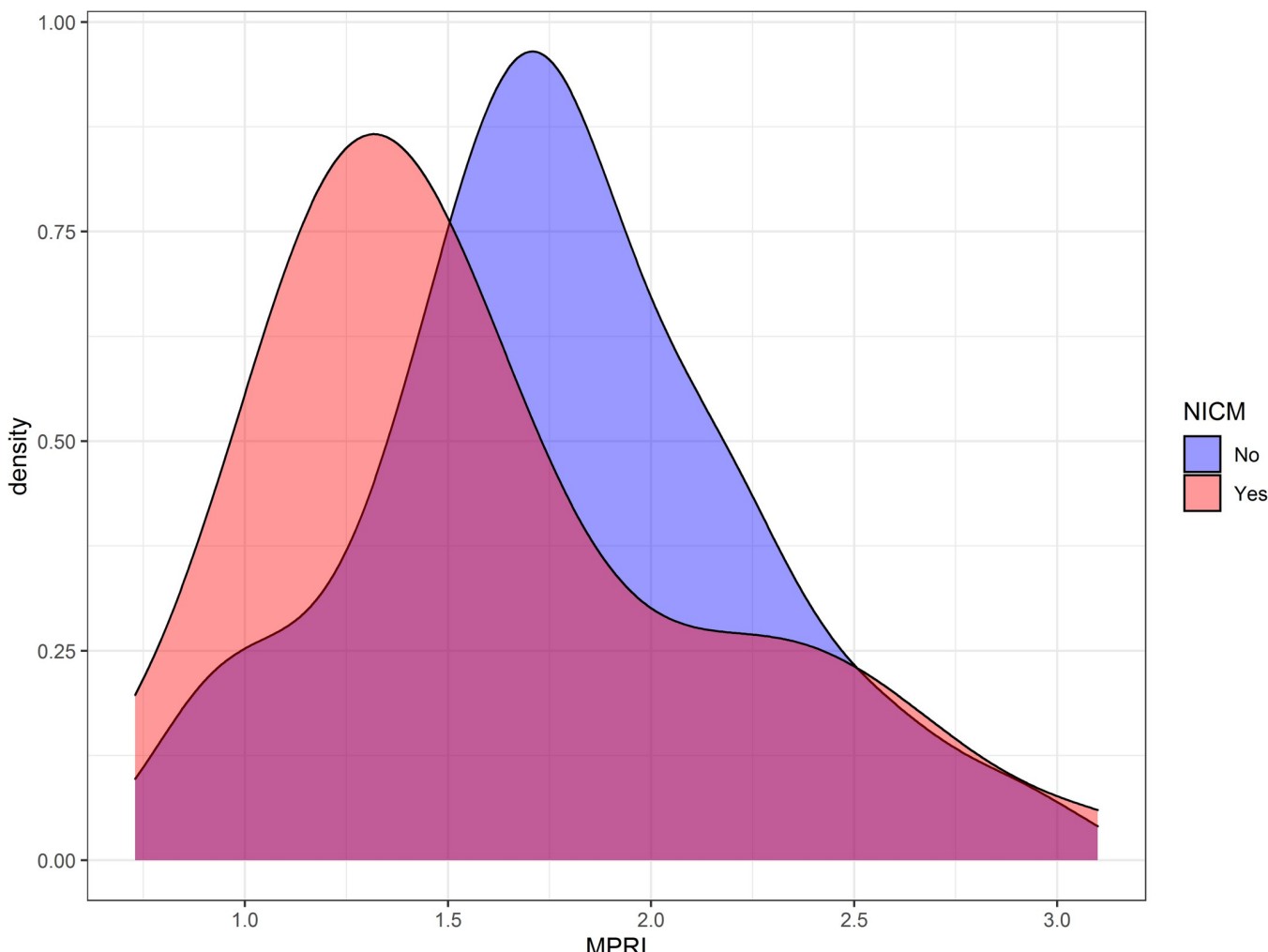

**Fig 4. Density plot histogram displaying distributions of myocardial perfusion reserve index (MPRI) for in a cardiomyopathy (pink) and preserved LVEF (blue) patients.**

correlation with both invasive and PET-derived CFR [27, 28]. In comparison to quantitative perfusion which requires a dedicated imaging sequence, semiquantitative techniques can be performed retrospectively on nearly any stress perfusion sequence, potentially allowing for more widespread availability.

An important aspect of our study is the ability to adjust for clinical and CMR covariates given the known link between MVD and hypertension, diabetes, and gender [7, 9]. Given the theoretical potential of LGE to affect regional microvascular function, we felt it important to control for the presence of LGE in multivariable analysis. MPRI remained significantly different between NICM and preserved LVEF patients even after controlling for the presence of LGE. Additionally, we did not identify a significant relationship between MPR and LV mass, volume, LVEF, or wall thickness, suggesting that intergroup differences in MPRI are unlikely to be related to artifact related to partial volume effects. We also note this significant relationship between hypertension and MVD. Due to its ability to evaluate for a multitude of etiologies of cardiomyopathy, CMR is often appropriately obtained early in the evaluation of newly diagnosed systolic heart failure [15]. CMR is therefore an ideal non-invasive test to assess microvasculature in this patient population. Unsurprisingly, rates of heart failure medication

**Table 2. Clinical and CMR characteristics in NICM patients with vs. without impaired myocardial perfusion reserve index.**

| | | MPRI >1.51 (N = 22) | MPRI <1.51 (N = 30) | P-value |
|---|---|---|---|---|
| **Clinical Characteristics** | | | | |
| Age (years) | | 52 (33–61) | 57 (38–63) | 0.37 |
| Gender, N (% male) | | 13 (72%) | 11 (48%) | 0.11 |
| Ethnicity | White, N (%) | 13 (72%) | 15 (65%) | 0.32 |
| | Black, N (%) | 4 (22%) | 8 (35%) | |
| | Hispanic, N (%) | 1 (6%) | 0 (0%) | |
| | Other, N (%) | 0 (0%) | 0 (0%) | |
| Diabetes, N (%) | | 2 (11%) | 4 (17%) | 0.57 |
| Hypertension, N (%) | | 7 (39%) | 17 (74%) | **0.02** |
| Hyperlipidemia, N (%) | | 6 (33%) | 10 (43%) | 0.51 |
| Atrial Fibrillation/Flutter, N (%) | | 1 (6%) | 2 (9%) | 0.70 |
| Creatinine (mg/dL) | | 0.90 (0.79–1.12) | 0.84 (0.75–1.05) | 0.67 |
| Hematocrit (%) | | 39 ± 7 | 37 ± 5 | 0.29 |
| B-type Natriuretic Peptide (ng/L) | | 206 (28–1500) | 69 (39–682) | 0.70 |
| Troponin-I (ng/mL) | | 0 (0–0.04) | 0.01 (0–0.05) | 0.61 |
| **CMR Characteristics** | | | | |
| LVEDD (cm) | | 5.7 (5.1–6.5) | 5.6 (5.3–6.2) | 0.75 |
| LVEDV (mL) | | 185 (158–226) | 177 (153–212) | 0.73 |
| LVEDV Indexed (mL/m$^2$) | | 100 (68–105) | 87 (70–108) | 0.89 |
| LVESV (mL) | | 113 (91–130) | 106 (79–152) | 1.0 |
| LVESV Indexed (mL/m$^2$) | | 55 (40–65) | 56 (41–76) | 0.77 |
| LVEF (%) | | 42 (38–46) | 43 (30–45) | 0.63 |
| LA Volume Indexed (gm/m$^2$) | | 43 (36–65) | 46 (44–58) | 0.63 |
| LV mass (gm) | | 109 (86–129) | 113 (83–135) | 0.77 |
| LV mass Indexed (gm/m$^2$) | | 49 (45–63) | 58 (46–64) | 0.38 |
| LGE presence (%) | | 11 (61%) | 10 (48%) | 0.26 |
| Number of segments | | 2 (0–3) | 0 (0–2) | 0.24 |

MPRI: myocardial perfusion reserve index; LV: left ventricle; EDD: end diastolic dimension; EDV: end diastolic volume; EF: ejection fraction; ESV: end systolic volume; LA: left atrium; LGE: late gadolinium enhancement.

**Table 3. Multivariable linear regression for predicting myocardial perfusion reserve.**

| | *B* coefficient (95% CI) | SE (*B*) | P value | 95% CI (bootstrap)* |
|---|---|---|---|---|
| NICM | -0.19 (-0.32 to -0.05) | 0.07 | 0.007 | -0.30 to -0.02 |
| Age | -0.005 (-0.009 to -0.0004) | 0.002 | 0.04 | -0.008 to 0.0005 |
| White Race | -0.06 (-0.21 to 0.09) | 0.08 | 0.43 | -0.22 to 0.08 |
| Male Gender | 0.09 (-0.04 to 0.22) | 0.06 | 0.15 | -0.03 to 0.24 |
| Diabetes | -0.11 (-0.29 to 0.07) | 0.09 | 0.24 | -0.26 to 0.14 |
| Hypertension | -0.15 (-0.27 to -0.03) | 0.06 | 0.02 | -0.29 to -0.04 |
| LGE | 0.10 (-0.053 to 0.239) | 0.07 | 0.19 | -0.11 to 0.22 |

NICM: non-ischemic cardiomyopathy; LGE: late gadolinium enhancement

Log-transformation for myocardial perfusion reserve was performed

* 2000 bootstrap sample

utilization—including ACE-inhibitors, angiotensin receptor blockers, and beta blockers—were higher in the NICM cohort; further studies are needed to better assess the impact of modern heart failure therapies on microvascular function in NICM.

Vatner et al investigated the presence of subendocardial ischemia in animal models of heart failure both at rest and with adenosine challenge [14]. These authors suggested that subendocardial ischemia may be a fundamental contributor to the progression of hypertensive heart disease to left ventricular systolic dysfunction. This proposed mechanism is supported by our finding that MVD was most prevalent in those with hypertension. Other investigators found progressive reduction in subendocardial blood flow with the evolution of ventricular systolic failure in a rapid pacing model and restoration of blood flow with normalization of ventricular function after cessation of pacing [13]. This suggests a contribution of MVD to the progression if not the early evolution of ventricular failure.

The presence of MVD may have important implications in patients with NICM. The presence of MVD has been previously associated with worse outcomes in this population [2]. Additionally, MVD may significantly contribute to exercise-related symptoms in patients with NICM. In comparison to controls, patients with microvascular angina exhibit greater impairments in diastolic function in response to adenosine [29]. Additionally, the presence of MVD is associated with impaired exercise tolerance in heart failure with preserved EF (HFpEF) patients [30]. This may explain in part why some heart failure patients experience symptoms with exercise in the absence of elevated filling pressures [31].

Improvements in subendocardial perfusion also appear can be seen in response to heart failure therapies and may parallel improvements in cardiac function. Leier et al reported a sustained improvement in ventricular function and symptoms following 72 hour infusions of dobutamine and suggested that the known impact of this positive inotropic agent on coronary blood flow may have corrected subendocardial ischemia and resulted in the observed benefits [32]. In support of this mechanism, similar improvements in ventricular function and symptoms were reported after 72 hour infusions of nitroglycerine [33]. Further research is needed to determine whether pharmacotherapies directly targeting the microvasculature are beneficial in the NICM population.

## Limitations

Our study is limited by its retrospective nature and therefore may be influenced by unknown confounders. The preserved LVEF group was identified retrospectively amongst patients referred for clinical CMR exam. Additionally, as patients in the preserved LVEF group were not excluded based on comorbidities, they cannot be viewed as a true "healthy control" group. However, this group is likely to be representative of the types of patients routinely seen in clinical practice. Additionally, because such patients were included, there were no significant intergroup differences in the prevalence of diabetes, atrial arrhythmias, and hyperlipidemia, thereby reducing the potential for confounding bias. We additionally attempted to address this by controlling for comorbidities in our multivariable analysis. The presence of significant MPRI differences despite matched MVD risk factors further underscores perfusion abnormalities in DCM, beyond what could be expected by traditional MVD risk factors. As this is a retrospective CMR-based study evaluating MPRI, invasive correlation was not obtained. Additionally, given the size of our cohort, we were unable to assess whether the presence of MVD derived from stress CMR was associated with adverse outcomes as has been shown previously with cardiac positron emission tomography (PET) [2]. Further studies are needed involving larger cohorts to determine whether stress CMR-derived semiquantitative perfusion is similarly prognostic.

## Conclusion

In conclusion, MVD—as assessed semi-quantitatively with CMR—is highly prevalent amongst patients with NICM as compared to subjects with preserved systolic function even after controlling for clinical and CMR covariates. Semiquantitative MPRI assessment detects MVD in significantly more patients as compared to visual evaluation alone. The expansion of stress CMR to include the evaluation of MVD will allow for improved phenotyping of NICM patients which may lead to better tailored therapies.

## Supporting information

**S1 Table. Comparison of rates of medical therapy utilization in NICM patients vs those with preserved LVEF.**
(DOCX)

**S2 Table. Multivariable gamma regression for predicting myocardial perfusion reserve.**
(DOCX)

**S1 Graphical abstract. Depicting the study design and main findings. Left panel:** study design and methods utilized to semi-quantitatively and visually assess microvascular disease (MVD) amongst patients with non-ischemic cardiomyopathy (NICM) and subjects with preserved left ventricular ejection fraction (LVEF). **Right panel, Main findings:** 1. Myocardial perfusion reserve index was significantly more impaired in NICM as compared to those with preserved LVEF. 2. Amongst those with NICM, MVD is detected significantly more often by semiquantitative as compared to visual methods.
(TIF)

## Acknowledgments

This original research was previously presented at the Society of Cardiovascular Magnetic Resonance (SCMR) 2020 Scientific Sessions and the Ohio State University Department of Heart and Lung Research Institute Annual Research Day in 2020. The authors do not wish to recognize any additional persons or institutions.

## Author Contributions

**Conceptualization:** Jeremy A. Slivnick, Karolina M. Zareba, Wojciech Mazur, Philip Binkley.

**Data curation:** Jeremy A. Slivnick, Vien T. Truong, Ellen Liu, Alexis Barnes.

**Formal analysis:** Jeremy A. Slivnick, Vien T. Truong, Philip Binkley.

**Investigation:** Jeremy A. Slivnick, Karolina M. Zareba, Philip Binkley.

**Methodology:** Jeremy A. Slivnick, Karolina M. Zareba, Wojciech Mazur, Philip Binkley.

**Supervision:** Philip Binkley.

**Writing – original draft:** Jeremy A. Slivnick.

**Writing – review & editing:** Karolina M. Zareba, Wojciech Mazur, Philip Binkley.

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
