## [Decision Letter · Decision Letter 0]

16 May 2022

PONE-D-22-03577Impairment in Quantitative Microvascular Function in Non-Ischemic Cardiomyopathy as Demonstrated Using Cardiovascular Magnetic ResonancePLOS ONE

Dear Dr. Slivnick,

Thank you for submitting your manuscript to PLOS ONE. After careful consideration, we feel that it has merit but does not fully meet PLOS ONE’s publication criteria as it currently stands. Therefore, we invite you to submit a revised version of the manuscript that addresses the points raised during the review process.

ACADEMIC EDITOR: All issues raised by expert reviewers are required.

We look forward to receiving your revised manuscript.

Kind regards,

Vincenzo Lionetti, M.D., PhD

Academic Editor

PLOS ONE

Journal Requirements:

Reviewers' comments:

Reviewer's Responses to Questions

**Comments to the Author**

1. Is the manuscript technically sound, and do the data support the conclusions?

Reviewer #1: Partly

Reviewer #2: Yes

Reviewer #3: Yes

2. Has the statistical analysis been performed appropriately and rigorously? 

Reviewer #1: Yes

Reviewer #2: Yes

Reviewer #3: Yes

3. Have the authors made all data underlying the findings in their manuscript fully available?

Reviewer #1: Yes

Reviewer #2: Yes

Reviewer #3: Yes

4. Is the manuscript presented in an intelligible fashion and written in standard English?

Reviewer #1: Yes

Reviewer #2: Yes

Reviewer #3: Yes

5. Review Comments to the Author

Reviewer #1: The Authors assessed the prevalence of MVD in 41 NICM pts as compared with 58 controls, using stress CMR. MVD was defined visually as the presence of a subendocardial perfusion defect and semiquantitatively by myocardial perfusion reserve index (MPRI<1.51). NICM patients had lower rest slope (3.9 vs 4.9, p=0.05), lower stress perfusion slope (8.8 vs 11.7, p<0.001), and MPRI (1.41 vs 1.74, p=0.02). Visual analysis had a lower sensitivity in detecting MVD. The topic is interesting, but several points should be addressed.

-One of the main limitation is the choice of control subjects, which appears as an heterogeneous group referred to CMR for several different reasons, and found (a posteriori) to have preserved LVEF (but 50-55% LVEF was included!), no LGE, no regional perfusion defects, no history of cardiac diseases. This is not a proper control group.

-In the control cohort, the indication for stress CMR was chest pain in 38 (66%), but it is unclear whether CAD was ruled out in these subjects. They still might suffer from microvascular angina, meaning that their MPRI is low because of an underlying cardiological disease.

-In the control patients, focal perfusion defects on first pass stress perfusion was an exclusion criterion, which is quite unclear. How many patients had an isolated perfusion defects?

-In the control patients, LGE was present in 16% of cases. What was the underlying disease? Might this impact on the microvascular function of these "normal" patients?

-In the methods, it is unclear whether a single short axis or 3 short axis slices were used for qualitative and semiquantitative perfusion analysis.

-A second main limitation is the lack of clinical and prognostic data. Actually, the Authors described MVD in two cohorts of patients with two different methods, without any reference standard, any correlatoion with disease type and severity (biohumoral data? arrhythmic burden?) or prognosis. Please comment/extend your data.

-In the conclusion, the Authors state that "Compared with visual analysis, semiquantitative analysis had a higher sensitivity for detecting impaired stress perfusion with CMR", but these thwo methods were not compared against a reference standard (PET? quantitative perfusion CMR?) to assess their real diagnostic accuracy. The Authors should revise this point, for example: "MVD by visual analysis was present in significantly fewer patients than MVD according to semiquantitative analysis...".

-A possible confounding variable is represented by the relatively thinner ventricles in NICM patients, which might increase the incidence of dark rim artifacts and should be carefully excluded from analysis. Did the Authors check for this point? Was there a relationship between wall thickness and MVD? And between LV volume and MVD?

Reviewer #2: The Authors retrospectively studied patients with non ischemic cardiomyopathy (NICM) and 58 control patients with preserved systolic function who underwent stress cardiac magnetic resonance (CMR) between 2011-2016. Microvascular disfunction (MVD) was defined visually as presence of a subendocardial perfusion defect and semiquantitatively by myocardial perfusion reserve index (MPRI). MPRI was compared between groups using univariate analysis and multivariable linear regression. 41 patients with NICM (mean age 51 ± 14, 59% male) and 58 controls (mean age 51 ± 13, 31% 35 male) were identified. In the NICM group, MVD was present in 23 (56%) and 11 (27%) by semiquantitative and visual evaluation respectively. Compared with controls, NICM patients had lower rest slope (3.9 vs 4.9, p=0.05) and stress perfusion slope (8.8 vs 11.7, p <0.001) and MPRI (1.41 vs 1.74, p=0.02). MPRI remained associated with NICM after controlling for several parameters, such as gender, hypertension, ethnicity, diabetes, and late gadolinium enhancement. The Authors observed that MVD—as assessed using CMR—is highly prevalent in NICM when compared to control patients with preserved systolic function. They concluded that semiquantitative assessment is more sensitive for detecting MVD compared to visual methods alone. The study was well planned and the results are really interesting, The method is suggesting and promising, stimulating further studies for the assessment of MVD.

Minor concerns are related to statistical analysis: please specify pair or unpair Student’s t-test.

Reviewer #3: This study uses CMR to assess coronary microvascular deficiency in NICM patients. It is well done,simple and clear.

There are some issues

1) Control group: it is not clear the control group population. It is very likely control group subjects were not healthy because, for example, in these patients LGE has been detected. The authors should select healthy subjects, almost without previous cardiovascular events, in the control group. Moreover, as the author stated, with regard the control group (line 91-92): Additionally, patients with infarct scar by LGE or focal perfusion defects on first pass stress perfusion were excluded from the control group. If we look at table, there are 9 subjects of the control group with LGE……….

2) In the multivariate analysis, the authors did not include any kind of functional and morphological cardiac parameter. It is important to know the relationship between MPRI and EF, MPRI and LV volumes. Also, it could be interesting to know the relationship between regional wall motion analysis (WMSI) and MPRI. This is because WMSI is one of the main cardiac variables with prognostic relevance. The authors stated that low MORI is not related to other clinical variables, but it is very likely they are related to morphological and functional cardiac variables.

3) In the discussion, the second main result, that is the higher diagnostic performance to detect MPR deficiency with semiquantitative method in respect to qualitative method is well known and cannot be a salient result of the study. The authors can mention this result but without highlighting it.

4) In the table 1, please report the value range of the indexed LVEDV. It is very likely that all NICM were not dilated.

5) Table 2: in this table the data are referred to all population and not to NICM patients with and without MPRI <> 1,51. In the text the authors wrote that in the table 2 are reported data of NICM patients only. Please, clear this point. However, it is more important to know the difference among NICM patient with and without MPRI < 1,51.

6. PLOS authors have the option to publish the peer review history of their article (what does this mean?). If published, this will include your full peer review and any attached files.

Reviewer #1: No

Reviewer #2: No

Reviewer #3: **Yes: **Alessandro Pingitore

---

## [Author Response · Author response to Decision Letter 0]

11 Aug 2022

Response to Reviewers: We thank the Reviewers for their thoughtful review of our manuscript entitled “Impairment in Quantitative Microvascular Function in Non-Ischemic Cardiomyopathy as Demonstrated Using Cardiovascular Magnetic Resonance.” Please see below for our revised manuscript accompanied by a response to each of the Reviewers' suggestions. We appreciate the opportunity to further address their valuable feedback and look forward to the opportunity to further revise as necessary.

---

## [Decision Letter · Decision Letter 1]

22 Sep 2022

PONE-D-22-03577R1Impairment in Quantitative Microvascular Function in Non-Ischemic Cardiomyopathy as Demonstrated Using Cardiovascular Magnetic ResonancePLOS ONE

Dear Dr. Slivnick,

Thank you for submitting your manuscript to PLOS ONE. After careful consideration, we feel that it has merit but does not fully meet PLOS ONE’s publication criteria as it currently stands. Therefore, we invite you to submit a revised version of the manuscript that addresses the points raised during the review process.

ACADEMIC EDITOR: All issues raised by reviewer are required.

We look forward to receiving your revised manuscript.

Kind regards,

Vincenzo Lionetti, M.D., PhD

Academic Editor

PLOS ONE

Journal Requirements:

Reviewers' comments:

Reviewer's Responses to Questions

**Comments to the Author**

1. If the authors have adequately addressed your comments raised in a previous round of review and you feel that this manuscript is now acceptable for publication, you may indicate that here to bypass the “Comments to the Author” section, enter your conflict of interest statement in the “Confidential to Editor” section, and submit your "Accept" recommendation.

Reviewer #1: (No Response)

Reviewer #2: All comments have been addressed

2. Is the manuscript technically sound, and do the data support the conclusions?

Reviewer #1: Yes

Reviewer #2: Yes

3. Has the statistical analysis been performed appropriately and rigorously? 

Reviewer #1: Yes

Reviewer #2: Yes

4. Have the authors made all data underlying the findings in their manuscript fully available?

Reviewer #1: Yes

Reviewer #2: Yes

5. Is the manuscript presented in an intelligible fashion and written in standard English?

Reviewer #1: Yes

Reviewer #2: Yes

6. Review Comments to the Author

Reviewer #1: The Authors performed a qualitative and a semiquantitative analysis of myocardial perfusion in 41 DCM patients compared with 58 patients with preserved EF. The manuscript is interesting and overall well written, even though there are some points to address:

-In patients with preserved EF (16% of which presented LGE), the final diagnosis was not reported; please include in the text or in a table (how many normal scans? how many prior myocarditis? other diseases?)

-Was severe valvular heart disease an exclusion criterion? What about mitral valve prolapse? Please specify in the methods, or comment in the results if any of the patients had similar diseases (which might impact on myocardial perfusion).

-There are no data about medical therapy, both in DCM and in patients with preserved EF; please include in a (supplemental?) table and comment in the text

-There is no mention of some recent studies on quantitative myocardial perfusion in DCM patients (See for example Gulati et al, DOI 10.1016/j.jcmg.2018.10.032), which should be mentioned and briefly discussed in the manuscript.

Reviewer #2: The paper has been revised and improved. It can be accepted for publication. The Authors discussed all the points of comments with success and accuracy.

7. PLOS authors have the option to publish the peer review history of their article (what does this mean?). If published, this will include your full peer review and any attached files.

Reviewer #1: No

Reviewer #2: No

---

## [Author Response · Author response to Decision Letter 1]

4 Oct 2022

Response to Reviewers: We thank the reviewer for his/her thoughtful review of our manuscript. We have strived to address the concerns raised by the reviewer regarding the manuscript; the comments have contributed to increased manuscript clarity and quality. Minor revisions were performed throughout the manuscript for typographical and grammatical errors. We appreciate the opportunity to address their valuable feedback and look forward to the opportunity to further revise as necessary.

Reviewer #1: The Authors performed a qualitative and a semiquantitative analysis of myocardial perfusion in 41 DCM patients compared with 58 patients with preserved EF. The manuscript is interesting and overall well written, even though there are some points to address:

We thank the Reviewer for his/her positive view of our work and the thoughtful comments and suggestions.

-In patients with preserved EF (16% of which presented LGE), the final diagnosis was not reported; please include in the text or in a table (how many normal scans? how many prior myocarditis? other diseases?)

We thank the reviewer for this advice. To better clarify this, we have added the following sentence (lines 166-168) “Within the NICM group, the etiology of cardiomyopathy was secondary to drug/toxin, genetic, hypertension, myocarditis, sarcoidosis, other, and idiopathic in 8 (20%), 3 (7%), 2 (5%), 2 (5%), 2(5%), 5 (12%), and 19 (41%) respectively (Figure 2).” Additionally, this information is also displayed in Figure 2 of the manuscript.

Was severe valvular heart disease an exclusion criterion? What about mitral valve prolapse? Please specify in the methods, or comment in the results if any of the patients had similar diseases (which might impact on myocardial perfusion).

We thank the reviewer for this valuable feedback. As the reviewer suggests, we did exclude those with severe valvular regurgitation or stenosis were excluded from both cohorts. To clarify this in the manuscript, the following statement has been added (lines 91-93), “Those with evidence of infarct scar (i.e. subendocardial or transmural) on CMR and those with severe valvular regurgitation or stenosis were also excluded from both cohorts.”

-There are no data about medical therapy, both in DCM and in patients with preserved EF; please include in a (supplemental?) table and comment in the text

The reviewer makes an excellent suggestion. Based on this feedback, we have gone back and collected rates of medical therapy between the two groups. Unsurprisingly, there was a higher rate of beta blocker and ACE/ARB utilization in the NICM group as compared to controls with LVEF. As suggested we have added a supplementary Table 1 describing these findings in the results section. Additionally, we have added the following sentence to the discussion section (lines 266-268) “Unsurprisingly, rates of heart failure medication utilization—including ACE-inhibitors, angiotensin receptor blockers, and beta blockers—were higher in the NICM cohort; further studies are needed to better assess the impact of modern heart failure therapies on microvascular function in NICM.”

-There is no mention of some recent studies on quantitative myocardial perfusion in DCM patients (See for example Gulati et al, DOI 10.1016/j.jcmg.2018.10.032), which should be mentioned and briefly discussed in the manuscript.

We thank the reviewer for sending us this excellent resource. In response, we have added a new paragraph to the discussion comparing our studies. Lines 250-256 now state, “Our study parallels a growing body of evidence including most notably Gulati et al in demonstrating a strong link between NICM and MVD2,25,26. A notable strength of the Gulati et al study was the utilization of fully quantitative CMR perfusion26. Fully quantitative perfusion has the potential to offer highly accurate quantification of myocardial blood flow at both stress and rest with robust correlation with both invasive and PET-derived CFR27,28. In comparison to quantitative perfusion which requires a dedicated imaging sequence, semiquantitative techniques can be performed retrospectively on nearly any stress perfusion sequence, potentially allowing for more widespread availability.”

---

## [Decision Letter · Decision Letter 2]

25 Oct 2022

PONE-D-22-03577R2Impairment in Quantitative Microvascular Function in Non-Ischemic Cardiomyopathy as Demonstrated Using Cardiovascular Magnetic ResonancePLOS ONE

Dear Dr. Slivnick,

Thank you for submitting your manuscript to PLOS ONE. After careful consideration, we feel that it has merit but does not fully meet PLOS ONE’s publication criteria as it currently stands. Therefore, we invite you to submit a revised version of the manuscript that addresses the points raised during the review process.

ACADEMIC EDITOR: Relevant issues addressed by one reviewer require a careful revision.

We look forward to receiving your revised manuscript.

Kind regards,

Vincenzo Lionetti, M.D., PhD

Academic Editor

PLOS ONE

Journal Requirements:

Reviewers' comments:

Reviewer's Responses to Questions

**Comments to the Author**

1. If the authors have adequately addressed your comments raised in a previous round of review and you feel that this manuscript is now acceptable for publication, you may indicate that here to bypass the “Comments to the Author” section, enter your conflict of interest statement in the “Confidential to Editor” section, and submit your "Accept" recommendation.

Reviewer #1: (No Response)

Reviewer #2: All comments have been addressed

2. Is the manuscript technically sound, and do the data support the conclusions?

Reviewer #1: Yes

Reviewer #2: Yes

3. Has the statistical analysis been performed appropriately and rigorously? 

Reviewer #1: Yes

Reviewer #2: Yes

4. Have the authors made all data underlying the findings in their manuscript fully available?

Reviewer #1: Yes

Reviewer #2: Yes

5. Is the manuscript presented in an intelligible fashion and written in standard English?

Reviewer #1: Yes

Reviewer #2: Yes

6. Review Comments to the Author

Reviewer #1: The Authors addressed the comments and improved the manuscript; as highlighted in a previous comment, I would suggest the Authors to better specify the final diagnosis not only of the 41 NICM patients (which has now been provided), but also of the 58 control patients (16% of which presented LGE: how many normal scans? how many prior myocarditis? other diseases in the control patients?).

Reviewer #2: The manuscript has been improved. The Authors did reply to the observations raised by referees, The manuscript can be accepted.

7. PLOS authors have the option to publish the peer review history of their article (what does this mean?). If published, this will include your full peer review and any attached files.

Reviewer #1: No

Reviewer #2: No

---

## [Author Response · Author response to Decision Letter 2]

30 Oct 2022

Response to Reviewers: We thank the reviewer for his/her thoughtful review of our manuscript. We have strived to address the concerns raised by the reviewer regarding the manuscript; the comments have contributed to increased manuscript clarity and quality. Minor revisions were performed throughout the manuscript for typographical and grammatical errors. We appreciate the opportunity to address their valuable feedback and look forward to the opportunity to further revise as necessary.

Reviewer #1: 

Comment 1: The Authors addressed the comments and improved the manuscript; as highlighted in a previous comment, I would suggest the Authors to better specify the final diagnosis not only of the 41 NICM patients (which has now been provided), but also of the 58 control patients (16% of which presented LGE: how many normal scans? how many prior myocarditis? other diseases in the control patients?).

Response: We thank the reviewer for this feedback. We have added the following sentence to the methods section (lines 96-97), “Additionally, those with disease-specific non-ischemic LGE patterns—such as myocarditis, sarcoidosis, or arrhythmogenic cardiomyopathy—were excluded from the preserved LVEF cohort.” Additionally, we have revised the results as follows (lines 186-188), “Among those in the preserved LVEF cohort in whom LGE was present, LGE was located in the insertion point in 7 (78%) patients; the remaining 2 (22%) patients had faint, non-specific inferolateral LGE.”

---

## [Decision Letter · Decision Letter 3]

8 Nov 2022

Impairment in Quantitative Microvascular Function in Non-Ischemic Cardiomyopathy as Demonstrated Using Cardiovascular Magnetic Resonance

PONE-D-22-03577R3

Dear Dr. Slivnick,

We’re pleased to inform you that your manuscript has been judged scientifically suitable for publication and will be formally accepted for publication once it meets all outstanding technical requirements.

Kind regards,

Vincenzo Lionetti, M.D., PhD

Academic Editor

PLOS ONE

Additional Editor Comments (optional):

Reviewers' comments:

Reviewer's Responses to Questions

**Comments to the Author**

1. If the authors have adequately addressed your comments raised in a previous round of review and you feel that this manuscript is now acceptable for publication, you may indicate that here to bypass the “Comments to the Author” section, enter your conflict of interest statement in the “Confidential to Editor” section, and submit your "Accept" recommendation.

Reviewer #1: All comments have been addressed

2. Is the manuscript technically sound, and do the data support the conclusions?

Reviewer #1: Yes

3. Has the statistical analysis been performed appropriately and rigorously? 

Reviewer #1: Yes

4. Have the authors made all data underlying the findings in their manuscript fully available?

Reviewer #1: Yes

5. Is the manuscript presented in an intelligible fashion and written in standard English?

Reviewer #1: Yes

6. Review Comments to the Author

Reviewer #1: (No Response)

7. PLOS authors have the option to publish the peer review history of their article (what does this mean?). If published, this will include your full peer review and any attached files.

Reviewer #1: No

---

## [Editor Report · Acceptance letter]

10 Nov 2022

PONE-D-22-03577R3 

Impairment in Quantitative Microvascular Function in Non-Ischemic Cardiomyopathy as Demonstrated Using Cardiovascular Magnetic Resonance 

Dear Dr. Slivnick:

I'm pleased to inform you that your manuscript has been deemed suitable for publication in PLOS ONE. Congratulations! Your manuscript is now with our production department. 

Kind regards, 

on behalf of

Prof. Vincenzo Lionetti 

Academic Editor

PLOS ONE